# Drifting codes within a stable coding scheme for working memory

**Michael J. Wolff** [ID] [1,2], **Janina Jochim** [3], **Elkan G. Akyürek** [2], **Timothy J. Buschman** [ID] [4], **Mark G. Stokes** [ID] [1,3] *

**1** Department of Experimental Psychology, University of Oxford, Oxford, United Kingdom, **2** Department of Experimental Psychology, University of Groningen, Groningen, the Netherlands, **3** Oxford Centre for Human Brain Activity, University of Oxford, Oxford, United Kingdom, **4** Princeton Neuroscience Institute and Department of Psychology, Princeton University, Princeton, New Jersey, United States of America

* mark.stokes@psy.ox.ac.uk

**Data Availability Statement:** All data and custom Matlab scripts used to generate the results and figures of this manuscript are available from the OSF database (osf.io/cn8zf).

## Abstract

Working memory (WM) is important to maintain information over short time periods to provide some stability in a constantly changing environment. However, brain activity is inherently dynamic, raising a challenge for maintaining stable mental states. To investigate the relationship between WM stability and neural dynamics, we used electroencephalography to measure the neural response to impulse stimuli during a WM delay. Multivariate pattern analysis revealed representations were both stable and dynamic: there was a clear difference in neural states between time-specific impulse responses, reflecting dynamic changes, yet the coding scheme for memorised orientations was stable. This suggests that a stable subcomponent in WM enables stable maintenance within a dynamic system. A stable coding scheme simplifies readout for WM-guided behaviour, whereas the low-dimensional dynamic component could provide additional temporal information. Despite having a stable subspace, WM is clearly not perfect—memory performance still degrades over time. Indeed, we find that even within the stable coding scheme, memories drift during maintenance. When averaged across trials, such drift contributes to the width of the error distribution.

## Introduction

Neural activity is highly dynamic, yet often we need to hold information in mind in a stable state to guide ongoing behaviour. Working memory (WM) is a core cognitive function that provides a stable platform for guiding behaviour according to time-extended goals; however, it remains unclear how such stable cognitive states emerge from a dynamic neural system.

At one extreme, WM could effectively pause the inherent dynamics by falling into a stable attractor (e.g., [1,2]). This solution has been well studied and provides a simple readout of memory content irrespective of time (i.e., memory delay). However, more dynamic models have also been suggested. For example, in a recent hybrid model, stable attractor dynamics coexist with a low-dimensional, time-varying component ([3,4], see Fig 1A for model schematics). This permits some dynamic activity whilst also maintaining a fixed coding relationship of

**Funding:** This research was in part funded by a James S. McDonnell Foundation Scholar Award (220020405) to MGS and an Open Research Area grant to MGS (ESRC ES/S015477/1) and EGA (NWO 464.18.114). The Wellcome Centre for Integrative Neuroimaging is supported by core funding from the Wellcome Trust (203139/Z/16/Z). The views expressed are those of the authors and not necessarily those of the National Health Service, the National Institute for Health Research or the Department of Health. The funders had no role in study design, data collection and analysis, decision to publish, or preparation of the manuscript.

**Competing interests:** The authors have declared that no competing interests exist.

**Abbreviations:** a.u., arbitrary units; CCW, counterclockwise; CW, clockwise; EEG, electroencephalography; fMRI, functional MRI; MDS, multidimensional scaling; norm. volt., normalised voltage; PFC, prefrontal cortex; WM, working memory.

WM content over time [5]. As in the original stable attractor model, the coding scheme is stable over time, permitting easy and unambiguous WM readout by downstream systems, regardless of maintenance duration [6]. Finally, it is also possible to maintain stable information in a richer dynamical system (e.g., [7]). Although the relationship between activity pattern and memory content changes over time, the representational geometry could remain relatively constant [5]. Such dynamics emerge naturally in a recurrent network and provide rich information about the previous input and elapsed time [8] but necessarily entail a more complex readout strategy (i.e., time-specific decoders or a high-dimensional classifier that finds a high-dimensional hyperplane that separates memory condition for all time points [9]).

Although all models seek to account for stable WM representations, it is also important to note that maintenance in WM is far from perfect. In particular, WM performance decreases over time [10], which could be ascribed to two different mechanisms (Fig 1B). On the one hand, the neural representation could degrade over time, either because of a decrease in WM-specific neural activity or through a broadening of the neural representation [11]. In this framework, the distribution of recall error reflects sampling from a broad underlying distribution. On the other hand, the neural representation of WM content might gradually drift along the feature dimension as a result of the accumulating effect of random shifts due to noise [12]. Even if the underlying neural representation remains sharp, variance in the mean over trials results in a relative broad distribution of errors over trials.

Computational modelling based on behavioural recall errors from WM tasks with varying set sizes and maintenance periods predicts a drift for colours and orientations maintained in WM [13,14]. At the neural level, evidence for drift has been found in the neural population code in monkey prefrontal cortex (PFC) during a spatial WM task [15], in which trial-wise shifts in the neural tuning profile predicted whether recall error was clockwise (CW) or counterclockwise (CCW) relative to the correct location. Recently, a human functional MRI (fMRI) study has found that delay activity reflected the probe stimulus more when participants erroneously concluded that it matched the memory item [16], which is consistent with the drift account.

Tracking these neural dynamics of nonspatial neural representations, which are not related to spatial attention or motor planning, is not trivial in humans. Previously, we found that the presentation of a simple impulse stimulus (task-relevant visual input) presented during the maintenance period of visual information in WM results in a neural response that reflects nonspatial WM content [17,18]. Here, we extend this approach to track WM dynamics. In the current study, we developed a paradigm to test the stability (and/or dynamics) of WM neural states and the consequence for readout by 'pinging' the neural representation of orientations at specific time points during maintenance.

We found that the coding scheme remained stable during the maintenance period, even though maintenance time was coded in an additional low-dimensional axis. We furthermore found that the neural representation of orientation drifts in WM. This was reflected in a shift of the reconstructed orientation towards the end of the maintenance period that correlated with behaviour.

## Results

In the present study, human participants completed a free-recall WM task while electroencephalography (EEG) was recorded (Fig 2). Visual impulses were presented at specific time points during WM maintenance, allowing us to track the neural dynamics of WM representations throughout the delay.

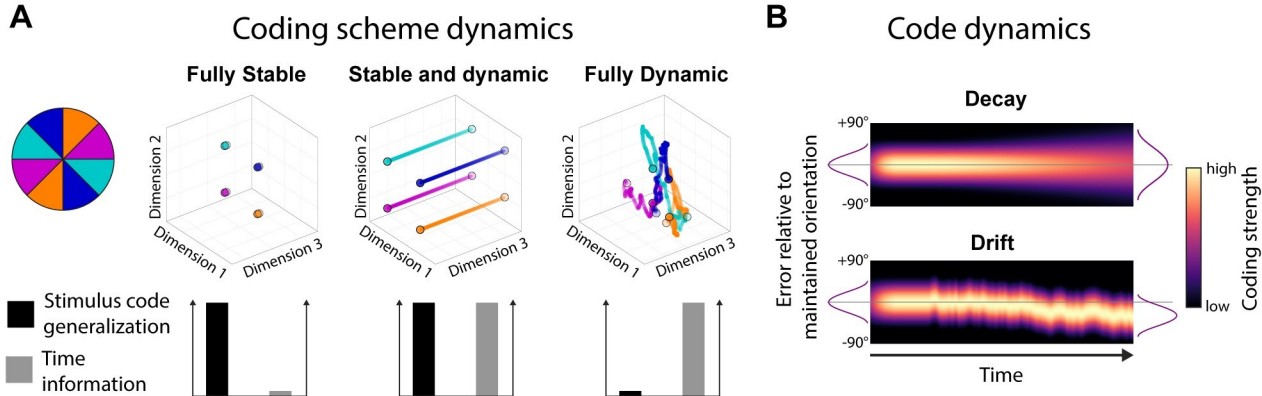

**Fig 1. Model predictions.** (A) The relationship between the neural coding scheme of orientations (colours) in WM over time, illustrated in neural state space (reduced to three dimensions, for visualisation). Left: A stable coding scheme within a stable neural population (defined by dimensions 1 and 2; dimension 3 has no meaningful variance). Middle: A stable coding scheme (dimensions 1 and 2) within a dynamic neural population (dimension 3). Right: A dynamically changing coding scheme (coding for orientation and time is mixed across dimensions). (B) The fidelity of the population code in WM over time. Top: The code decays and becomes less specific over time, leading to random errors during readout. Bottom: The code drifts along the feature dimension, leading to a still sharp, but shifted code during readout. WM, working memory.

## Item and WM content-specific evoked responses during encoding and maintenance

The neural response elicited by the memory array contained information about the presented orientations ($p < 0.001$, one-sided; Fig 3, left). The first impulse response contained statistically significant information about the cued item ($p = 0.011$, one-sided) but not the uncued item, which failed to reach the statistical significance threshold ($p = 0.051$, one-sided). The difference between cued and uncued item decoding was not significant ($p = 0.694$, two-sided; Fig 3, middle). The decodability of the cued item was also significant at the second impulse response ($p < 0.001$, one-sided), whereas it was not of the uncued item ($p = 0.921$, one-sided). The decodability of the cued item was significantly higher than that of the uncued item ($p = 0.002$, two-sided; Fig 3, right).

Overall, these results reflect previous findings [18] in that the impulse response reflects relevant information in WM. However, the marginally significant decoding of the uncued item at impulse 1 suggests that the item might not have been completely dropped from memory approximately 0.9 seconds after cue and 1.6 seconds before probe presentation. Nevertheless,

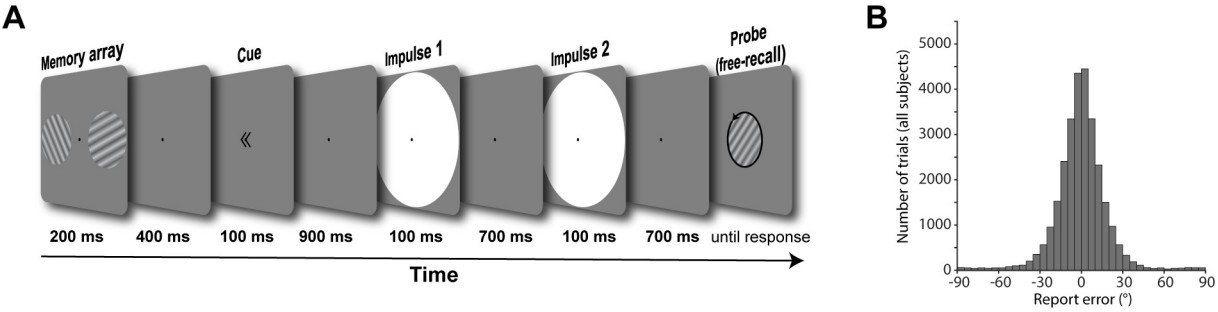

**Fig 2. Trial schematic and behavioural results.** (A) Two randomly orientated grating stimuli were presented laterally. A retro-cue then indicated which of those two would be tested at the end of the trial. Two impulses (white circles) were serially presented in the subsequent delay period. At the end of the trial, a randomly oriented probe grating was presented in the centre of the screen, and participants were instructed to rotate this probe until it reflected the cued orientation. (B) Report errors of all trials across all participants. Data available at osf.io/cn8zf.

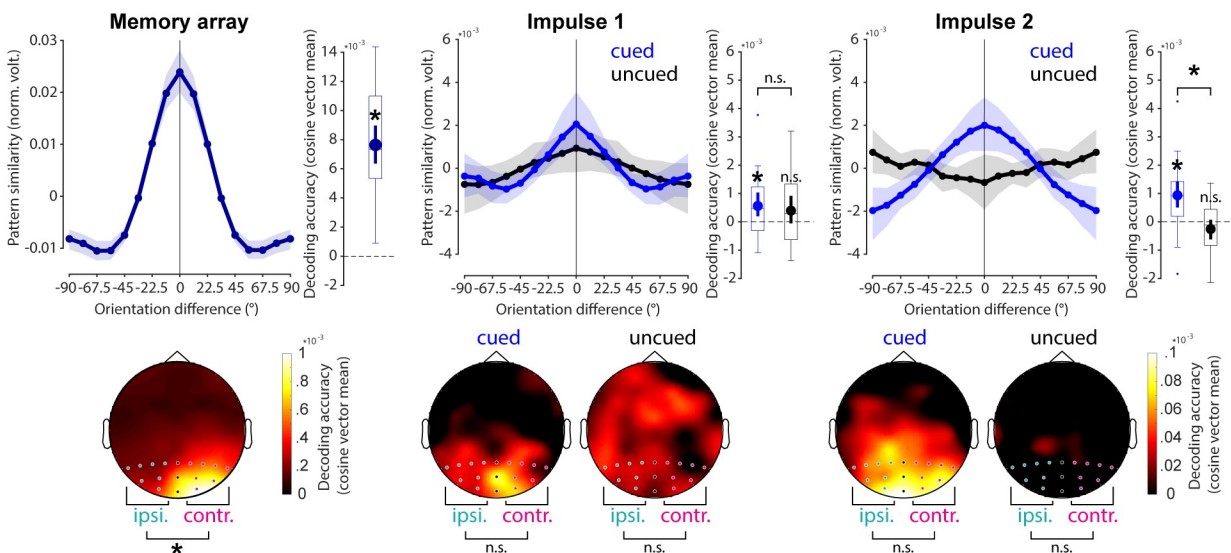

**Fig 3. Decoding results.** Top row: Normalised average pattern similarity (mean-centred, sign-reversed Mahalanobis distance) of the evoked neural responses (100 to 400 ms relative to stimulus onset) as a function of orientation similarity, and decoding accuracy (cosine vector means of pattern similarities). Error shadings and error bars are 95% CI of the mean. Asterisks indicate significant decoding accuracies ($p < 0.05$, one-sided) or differences ($p < 0.05$, two-sided). Bottom row: Decoding topographies of the searchlight analysis. Posterior channels used in all other decoding analyses are highlighted. Ipsilateral ('ipsi.') and contralateral ('contr.') channels used to test for item lateralisation are highlighted in turquoise and pink, respectively. Data available at osf.io/cn8zf. norm. volt., normalised voltage; n.s., not significant.

at impulse 2 (about 1.7 seconds after cue), no detectable trace of the uncued item remained, confirming that participants likely removed it from memory for optimal processing of the probe stimulus.

The decoding topographies highlight that most of the decodable signal came from posterior electrodes during both encoding and maintenance and is therefore likely generated by the visual cortex (Fig 3, bottom row). The decoding difference between contralateral and ipsilateral posterior electrodes (P7/8, P5/6, P3/4, P1/2, PO7/8, PO3/4, O1/2) was significantly different during item encoding, with higher item decoding at contralateral compared to ipsilateral electrodes ($p < 0.001$, two-sided). Interestingly, no evidence for such lateralisation was found at either impulse 1 (cued item: $p = 0.854$; uncued item: $p = 0.526$, two-sided) or impulse 2 (cued item: $p = 0.716$; uncued item: $p = 0.398$, two-sided).

## Stable WM coding scheme in time

The relationship between orientations and impulses/time is visualised in state space through multidimensional scaling (MDS; Fig 4A). Whereas the first dimension clearly differentiates between impulses, the second and third dimensions code the circular geometry of orientations in both impulses, suggesting that whereas the impulse responses are different between impulses, the orientation coding schemes revealed by the impulses are the same. This is corroborated by significant decoding accuracy of the impulses ($p < 0.001$, one-sided; Fig 4B) on the one hand but also significant cross-generalisation of the orientation code between impulses ($p < 0.001$, two-sided), which was not significantly different from same-impulse orientation decoding ($p = 0.608$, two-sided; Fig 4C).

For completeness, we also report the full cross-temporal generalisation matrix between impulses using a continuous decoding analysis (S1 Fig), in which a time-resolved classifier was trained and tested on all possible time point–by–time point combinations [19].

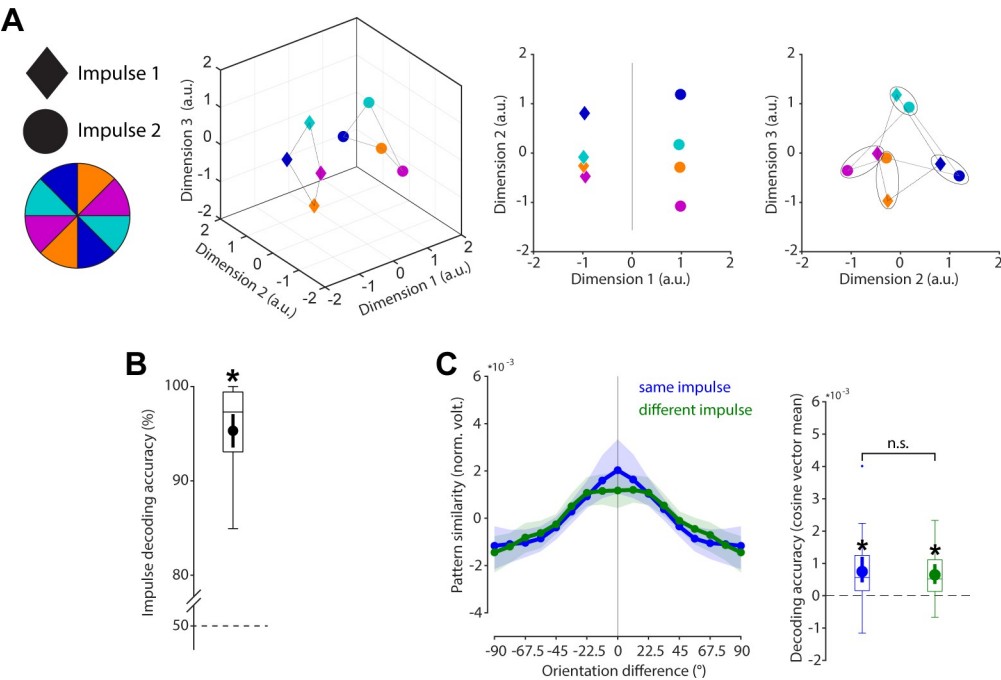

**Fig 4. Cross-generalisation of coding scheme between impulses.** (A) Visualisation of orientation and impulse code in state space. The first dimension discriminates between impulses. The second and third dimensions code the orientation space in both impulses. (B) Trial-wise accuracy (%) of impulse decoding. (C) Orientation decoding within each impulse (blue) and orientation code cross-generalisation between impulses (green). Error shadings and error bars are 95% CI of the mean. Asterisks indicate significant decoding accuracies or cross-generalisation ($p < 0.05$). Data available at osf.io/cn8zf. a.u., arbitrary units; norm. volt., normalised voltage; n.s., not significant.

To rule out that the difference in impulse response reported above (Fig 4B) is not only due to differences in stimulation history and changing WM operations but also due to temporal coding, we reanalysed previously published data in which a single impulse stimulus was presented either 1,170 or 1,230 ms after the presentation of a single memory item [17]. The findings largely replicate the results reported above: State-space visualisation of impulse onset and orientations shows the same circular geometry of the orientations at each impulse onset while also highlighting a separation of impulse onsets in state space (S2A Fig). Decoding impulse onset was significantly higher than chance ($p = 0.004$, one-sided; S2B Fig). Cross-generalisation of the orientation code between impulse onsets was significant ($p < 0.001$, two-sided) and did not significantly differ from decoding the memorised orientation within the same impulse onset ($p = 0.240$, two-sided; S2C Fig).

Overall, the results of the current study, as well as the reanalyses of [17], provide evidence for a low-dimensional change over time, which can be revealed by perturbing the WM network at different time points (as predicted in [20]), while at the same time providing evidence for a temporally stable coding scheme of WM content [3,4]. Note that a stable coding scheme at the global scale (as revealed by EEG in the present study) does not rule out the possible existence of WM-specific neurons that exhibit time-varying activity during WM maintenance [9,21].

## Specific WM coding scheme in space

As a counterpart to the stable coding scheme in time reported above, we explicitly tested whether the coding scheme is location specific (i.e., dependent on the previous presentation

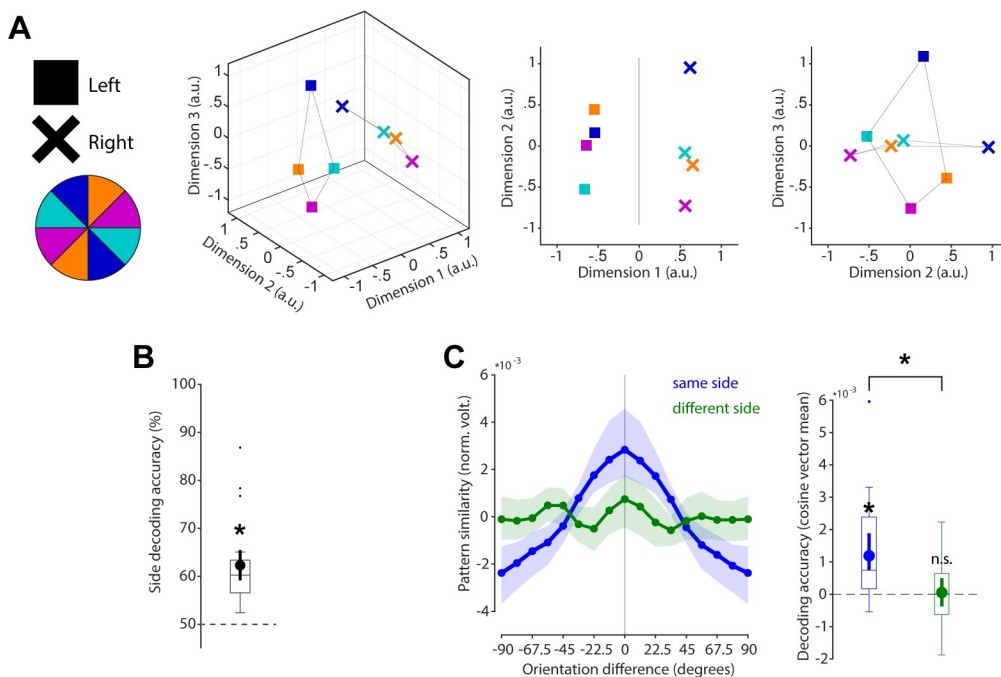

**Fig 5. No cross-generalisation of coding scheme between cued item locations during impulse responses.** (A) Visualisation of orientation and item location code in state space. The first dimension discriminates between item locations. The first and second dimensions code the orientation space, separately for WM items previously presented on the left or right side. (B) Trial-wise accuracy (%) of item location decoding. (C) Orientation decoding within each item location (blue) and orientation code cross-generalising between different item locations (green). Error shadings and error bars are 95% CI of the mean. Asterisks indicate significant decoding accuracies and differences ($p < 0.05$). Data available at osf.io/cn8zf. a.u., arbitrary units; norm. volt., normalised voltage; n.s., not significant; WM, working memory.

location of the cued orientation). State-space visualisation of cued item location and orientations shows a clear separation between locations and no overlap in orientation coding between locations (Fig 5A). The cued location was significantly decodable from the impulse responses ($p < 0.001$, one-sided; Fig 5B). Cross-generalisation of the orientation coding scheme between cued item locations was not significant ($p = 0.376$, two-sided) and was significantly lower than same-side orientation decoding ($p = 0.004$, two-sided; Fig 5C). These results reflect previous reports of spatially specific WM codes, even when location is no longer relevant [22], though we cannot rule out the presence of spatially invariant representations that are not detectable with our experiment.

## Drifting WM code

The first approach to test for a possible shift of the neural representation towards the adjusted response (i.e., without report bias, see Methods and S3 Fig) averaged the trial-wise orientation similarity profiles obtained from the cross-validated orientation reconstruction on all trials (see Methods and Fig 6A). No significant shift towards the response was evident during encoding/memory array presentation (circular mean: $p = 0.117$; asymmetry score: $p = 0.125$, one-sided; Fig 6B and 6C, left). No evidence for such a shift was found at impulse 1/early maintenance either (circular mean: $p = 0.07$; asymmetry score: $p = 0.057$, one-sided; Fig 6B and 6C, middle). However, the orientation similarity profile was significantly shifted towards the response at impulse 2/late maintenance (circular mean: $p < 0.001$; asymmetry score: $p < 0.001$, one-sided; Fig 6B and 6C, right).

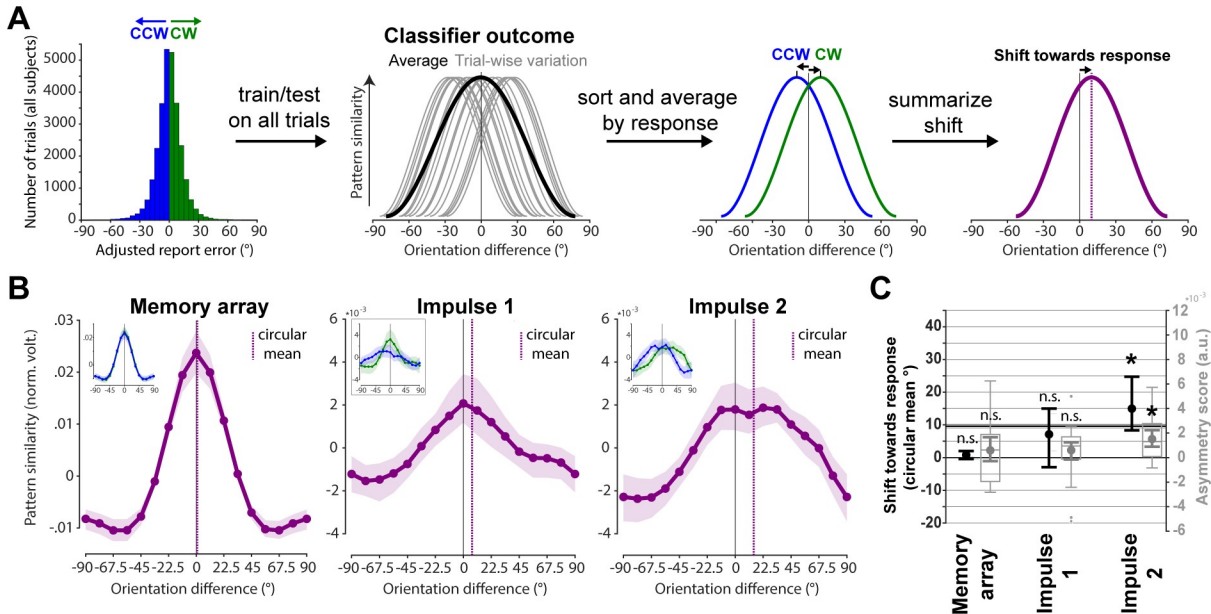

**Fig 6. Response-dependent averaging of trial-wise similarity profiles demonstrates drift. Schematic and results.** (A) Testing for shift towards response by averaging trial-wise similarity profiles by CCW/CW responses. (B) Results of schematised approach in (A). Orientation similarity profiles averaged by response such that a rightward shift reflects a shift towards the response (purple) at each event. Purple vertical lines show circular means of the similarity profiles. Insets show orientation similarity profiles for CCW (blue) and CW (green) responses separately. Error shadings are 95% CI of the mean. (C) Group-level (circular mean) and participant-level (asymmetry score) shifts towards the response of each response-dependent similarity profile are shown in black and grey, respectively. Error bars are 95% CI of the mean. The blue line and shading indicates the mean and 95% CI of the absolute, bias-adjusted behavioural response deviation (approximately 10 degrees). Data available at osf.io/cn8zf. a.u., arbitrary units; CCW, counterclockwise; CW, clockwise; norm. volt., normalised; n.s., not significant.

The second approach to test for a possible shift of the neural representation towards the adjusted response may be more sensitive, since it trains the orientation classifier only on CCW trials and tests it on CW trials, and vice versa (see Methods and Fig 7A), thus increasing any response related shift by a factor of two. This approach yielded similar results as the previous approach, though the shift magnitudes are indeed larger. Neither the memory array presentation/encoding (circular mean: $p = 0.124$; asymmetry score: $p = 0.129$, one-sided) nor impulse 1/early maintenance (circular mean: $p = 0.104$; asymmetry score: $p = 0.082$, one-sided) showed a significant shift towards the response (Fig 7B and 7C, left and middle), whereas impulse 2/ late maintenance did (circular mean: $p < 0.001$; asymmetry score: $p < 0.001$, one-sided; Fig 7B and 7C, right).

Note the reported results of shifts during impulse presentations were obtained by training the classifier on both impulses but testing it on each impulse separately. This was done to improve power (as explained in Methods). This improved orientation reconstruction, particularly for the latter shift analysis in which the classifier is trained on only half the trials (CW trials only or CCW trials only). However, the same analyses based on training (and testing) within each impulse epoch separately yielded qualitatively similar results (no significant shifts at impulse 1 in either approach, significant shifts at impulse 2 in both approaches; S4 Fig).

## Discussion

In the present study, we investigated the neural dynamics of WM by probing the coding scheme over time, as well as drift in the actual memories. The neural responses to impulse stimuli in this nonspatial WM paradigm enabled us to show that the coding scheme of a

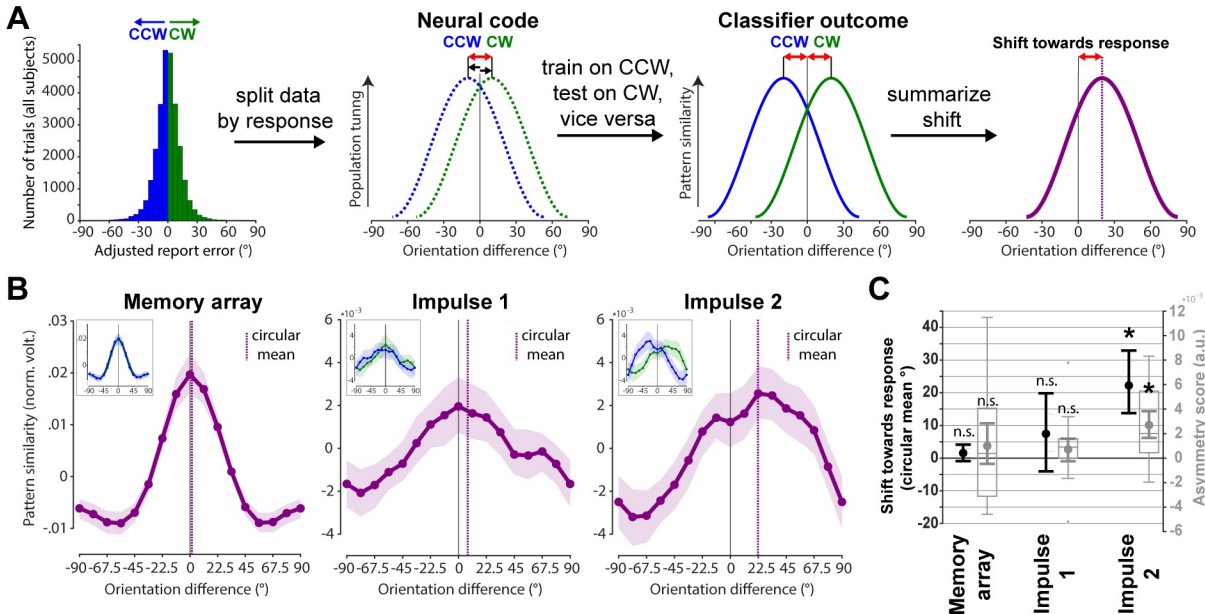

**Fig 7. Response-dependent training and testing demonstrates drift. Schematic and results.** (A) Testing for shift towards response by first splitting the neuroimaging data into CW and CCW data sets and training on CW trials and testing on CCW trials, and vice versa. Given an actual shift, the shift of the resulting orientation reconstruction will be doubled, since training and testing data are shifted in opposite directions. (B) Results of schematised approach in (A). Average orientation similarity profiles such that a rightward shift reflects a shift towards the response (purple) at each event. Purple vertical lines show circular means of the similarity profiles. Insets show orientation similarity profiles for CCW (blue) and CW (green) responses separately. Error shadings are 95% CI of the mean. (C) Group-level (circular mean) and participant-level (asymmetry score) shifts towards the response of each response-dependent similarity profile are shown in black and grey, respectively. Error bars are 95% CI of the mean. Data available at osf.io/cn8zf. a.u., arbitrary units; CCW, counterclockwise; CW, clockwise; norm. volt., normalised voltage; n.s., not significant.

parametric visual feature (i.e., orientation) in WM remained stable during maintenance, reflected in the significant cross-generalisation of the orientation decoding between early and late impulses (Fig 4). However, memories drift within this stable coding scheme, leading to a bias in memories (Figs 6 and 7).

This is consistent with previous reports of a stable subspace for WM maintenance [4,5] and provides evidence for a time-invariant coding scheme for orientations maintained in WM. However, more dynamic schemes have also been reported [23]—for example, during the early transition between encoding and maintenance [24,25]. At the extreme end, some have proposed that WM could be maintained in a dynamical system, in which activity continues to evolve throughout the delay period along a complex trajectory in neural state space (e.g., [26]), possibly through sequential activation of neurons (e.g., [27]). Dynamic trajectories emerge naturally from recurrent neural networks and provide additional information, such as elapsed time [28]. However, the dimensionality of dynamic coding places an important constraint on the generalisability of a particular coding scheme over time [6]. In the current study, we find evidence for a hybrid model [3,4]: stable decoding of WM content, despite dynamic activity over time.

Specifically, although there was no cost of cross-generalising the orientation code between impulses, there was a clear difference in the neural pattern between them, suggesting that a separate (low-dimensional) dynamic neural pattern codes the passage of time. A reanalysis of the data of a previously published study [17] confirmed these results, suggesting that the low-dimensional dynamics code for time per se (rather than impulse number). Importantly, the low-dimensional representation of elapsed time is orthogonal to the mnemonic subspace,

allowing WM representations to be stable. This hybrid of stable and dynamic representations may emerge from interactions between dynamic recurrent neural networks and stable sensory representations [3]. It is also possible that more complex dynamics could be observed in a more complex WM paradigm [23].

Our index of WM-related neural activity was based on an impulse response approach that we previously developed to measure WM-related changes in the functional state of the system [17,18], including 'activity-silent' WM states [29,30]. For example, activity states during encoding could result in a neural trace in the WM network through short-term synaptic plasticity [31,32], resulting in a stable code for maintenance, whereas the time dimension could be represented in its gradual fading [20,33,34]. The stable WM-content coding scheme could also be achieved by low-level activity states that self-sustain a stable code through recurrent connections, a key feature of attractor models of WM [1,35], whereas dynamic activity patterns are coded in an orthogonal subspace that represents time. Although we did not explicitly consider tonic delay activity, it is nonetheless possible that the impulse responses also reflect nonlinear interactions with low-level, persistent activity states that are otherwise difficult to measure with EEG. Therefore, we cannot rule out a contribution of persistent activity in the stable coding scheme observed here.

We also found evidence that the orientation code itself drifts along the orientation dimension, which is correlated with recall errors. Whereas there was no bias in the neural orientation representation at either encoding or early maintenance, the second impulse towards the end of the maintenance period revealed a code that was shifted towards the direction of response error. This pattern of results is consistent with the drift account of WM, in which neural noise leads to an accumulation of error during maintenance, resulting in a still sharp, but shifted (i.e., slightly wrong) neural representation of the maintained information [1,14]. Although previous neurophysiological recordings from monkey PFC found evidence for drift for spatial information [15], we could demonstrate a shifting representation that more faithfully represents nonspatial WM content that is unrelated to sustained spatial attention or motor preparation, by using lateralised orientations in the present study.

Bump attractors have been proposed as an ideal neural mechanism for the maintenance of continuous representations (i.e., space, orientation, colour), in which a specific feature is represented by the persistent activity 'bump' of the neural population at the feature's location along the network's continuous feature space. Neural noise randomly shifts this bump along the feature dimension, whereas inhibitory and excitatory connections maintain the same overall level of activity and shape of the neural network [36,37]. Random walk along the feature dimension is thus a fundamental property of bump attractors and has been found to explain neurophysiological findings [15]. Typically, this is considered within the framework of persistent WM; however, transient bursts of activity could also follow similar attractor dynamics [32,38]. For example, although temporary connectivity changes of the memorised WM item may indeed slowly dissolve and become coarser, periodic activity bursts may keep this to a minimum by periodically reinstating a sharp representation [32]. However, since this refreshing depends on the readout of a coarse representation, the resulting representation may be slightly wrong and thus shifted. This interplay between decaying silent WM states that are read out and refreshed by active WM states also predicts a drifting WM code, without depending on an unbroken chain of persistent neural activity.

Moreover, the representational drift does not necessarily have to be random. Modelling of report errors in a free-recall colour WM task suggests that an increase of report errors over time may be due to separable attractor dynamics, with a systematic drift towards stable colour representations, resulting in a clustering of reports around specific colour values, in addition to random drift elicited by neural noise [13]. The report bias of oblique orientations seen in

the present study could be explained by a similar drift towards specific orientations, which would predict an increase of report bias for longer retention periods. However, clear behavioural evidence for such an increase in systemic report errors of orientations is lacking [10]. In the present study, we isolated random from systematic errors, both as a methodological necessity and to allow us to attribute any observed shift to random errors. Thus, although a systematic drift towards specific orientations might be possible, the shift in representation reported here is unrelated to it.

Our results suggest that maintenance in WM is dynamic, although the fundamental coding scheme remains stable over time. Low-dimensional dynamics could provide a valuable readout of elapsed time whilst allowing for a time-general readout scheme for the WM content. We also show that drift within this stable coding scheme could explain loss of memory precision over time.

## Methods

### Ethics statement

The study was approved by the Central University Research Ethics Committee of the University of Oxford (R42977/RE001), which adheres to the Declaration of Helsinki. Participants gave written informed consent prior to participation.

### Participants

Twenty-six healthy adults (17 female, mean age 25.8 years, range 20–42 years) were included in all analyses. Four additional participants were excluded during preprocessing because of excessive eye movements (more than 30% of trials contaminated). Participants received monetary compensation (£10 an hour) for participation.

### Apparatus and stimuli

The experimental stimuli were generated and controlled by Psychtoolbox [39], a freely available MATLAB extension. Visual stimuli were presented on a 23-inch (58.42-cm) screen running at 100 Hz and a resolution of 1,920 by 1,080. Viewing distance was set at 64 cm. A Microsoft Xbox 360 controller was used for response input by the participants.

A grey background (RGB = 128, 128, 128; 20.5 cd/m$^2$) was maintained throughout the experiment. A black fixation dot with a white outline (0.242˚) was presented in the centre of the screen throughout all trials. Memory items and the probe were sine-wave gratings presented at 20% contrast, with a diameter of 8.51˚ and spatial frequency of 0.65 cycles per degree, with randomised phase within and across trials. Memory items were presented at 6.08˚ eccentricity. The rotation of memory items and probe were randomised individually for each trial. The impulse stimulus was a single white circle, with a diameter of 20.67˚, presented at the centre of the screen. The retro-cue was two arrowheads pointing right (>>) or left (<<) and was 1.58˚ wide. A coloured circle (3.4˚) was used for feedback. Its colour depended dynamically on the precision of recall, ranging from red (more than 45 degrees error) to green (0 degrees error). A pure tone also provided feedback on recall accuracy after each response, ranging from 200 Hz (more than 45 degrees error) to 1,100 Hz (0 degrees error).

### Procedure

Participants participated in a free-recall, retro-cue visual WM task. Each trial began with the fixation dot. Participants were instructed to maintain central fixation throughout each trial. After 1,000 ms, the memory array was presented for 200 ms. After a 400 ms delay, the retro-

cue was presented for 100 ms, indicating which of the previously presented items would be tested, rendering the other item irrelevant. The first impulse stimulus was presented for 100 ms, 900 ms after the offset of the retro-cue. After a delay of 700 ms, the second impulse stimulus was presented for 100 ms. After another delay of 700 ms, the probe was presented. Participants used the left joystick on the controller with the left thumb to rotate the orientation of the probe until it best reflected the memorised orientation and confirmed their answer by pressing the 'x' button on the controller with the right thumb. Note that one complete rotation of the joystick corresponded to 0.58 of a rotation of the probe. In conjunction with the fact that the probe was randomly orientated on each trial, it was impossible for participants to plan the rotation before-hand or memorise the direction of the joystick instead of the orientation of the memory item. Accuracy feedback was given immediately after the response in which both the coloured circle and tone were presented simultaneously. Each participant completed 1,100 trials in total, over a course of approximately 135 minutes, including breaks. See Fig 2A for a trial schematic.

## EEG acquisition

EEG was acquired with 61 Ag/AgCl sintered electrodes (EasyCap, Herrsching, Germany) laid out according to the extended international 10–20 system and recorded at 1,000 Hz using Curry 7 software (Compumedics NeuroScan, Charlotte, NC). The anterior midline frontal electrode (AFz) was used as the ground. Bipolar electrooculography (EOG) was recorded from electrodes placed above and below the right eye and the temples. The impedances were kept below 5 kΩ. The EEG was referenced to the right mastoid during acquisition.

## EEG preprocessing

Offline, the EEG signal was re-referenced to the average of both mastoids, down-sampled to 500 Hz, and bandpass filtered (0.1 Hz high-pass and 40 Hz low-pass) using EEGLAB [40]. The continuous data were epoched relative to the memory array onset (−500 ms to 3,600 ms) before independent component analysis [41] was applied. Components related to eye blinks were subsequently removed. The data were then epoched relative to memory array onset and the two impulse onsets (0 ms to 400 ms), and trials were individually inspected. Trials with loss of fixation, visually identified from the EOG, and trials with nonarchetypical artefacts, visually identified from the EEG, in the memory array epoch and in either impulse epoch were removed from all subsequent analyses. Furthermore, trials in which the report error was 3 circular standard deviations (SDs) from the participant's mean response error were also excluded from EEG analyses to remove trials that likely represent complete guesses [42]. This led to the removal of $M$ = 2.3% (SD = 1.2%) trials because of inaccurate report trials, in addition to the $M$ = 3.52% (SD = 4.21%) and $M$ = 5% (SD = 5.2%) of trials removed because of eye movements and nonarchetypical EEG artefacts from the memory array and impulse epochs, respectively.

MVPA on electrophysiological data is usually performed on each time point separately. However, taking advantage of the highly dynamic waveform of evoked responses in EEG by pooling information multivariately over electrodes as well as time can improve decoding accuracy, at the expense of temporal resolution [43,44]. Since the previously reported WM-dependent impulse response reflects the interaction of the WM state at the time of stimulation and does not reflect continuous delay activity, we treat the impulse responses as discrete events in the current study. Thus, the whole time window of interest relative to impulse onsets (100 to 400 ms) from the 17 posterior channels was included in the analysis. The time window was based on previous, time-resolved findings, which showed that the WM-dependent neural response from a 100-ms impulse (as used in the current study) is largely confined to this window [18]. In the current study, instead of decoding at each time point separately, information

was pooled across the whole time window. The mean activity level within each time window was first removed for each trial and channel separately, thus normalising the voltage fluctuations over time and isolating the dynamic, impulse-evoked neural signal from more stable brain states. The time window was then down-sampled to 100 Hz by taking the average every 10 ms. This was done to reduce the number of dimensions, which both reduces computational demands but also improves signal to noise by removing redundant dimensions of extremely high-frequency voltage changes in the EEG (>100 Hz) that are unlikely to reflect genuine brain activity. This resulted in 30 values per channel, each of which was treated as a separate dimension in the subsequent multivariate analysis (510 in total). This data format was used on all subsequent MVPA analyses, unless explicitly mentioned otherwise. The same approach over the same time window of interest was used in our previous study [45].

### Orientation reconstruction

We computed the Mahalanobis distances as a function of orientation difference to reconstruct grating orientations [18]. The following procedure was performed separately for items that were presented on the left and right side. Since the grating orientations were determined randomly on a trial-by-trial basis and the resulting orientation distribution across trials was unbalanced, we used a k-fold procedure with subsampling to ensure unbiased decoding. Trials were first assigned the closest of 16 orientations (variable, see below), which were then randomly split into 8 folds using stratified sampling. Using cross-validation, the train trials in 7 folds were used to compute the covariance matrix using a shrinkage estimator [46]. The number of trials of each orientation bin in the 7 train folds were equalised by randomly subsampling the minimum number of trials in any bin. The subsampled train trials of each angle bin were then averaged. To pool information across similar orientations, the average bins of the train trials were convolved with a half cosine basis set raised to the 15th power [47–49]. The Mahalanobis distances between each trial of the left-out test fold and the averaged and basis-weighted angle bins were computed. The resulting 16 distances per test trial were normalised by mean centring them. This was repeated for all test and train fold combinations. To get reliable estimates, the above procedure was repeated 100 times (random folds and subsamples each time), separately for 8 orientation spaces (0° to 168.75°, 1.40625° to 170.1563°, 2.8125° to 171.5625°, 4.2188° to 172.9688°, 5.625° to 174.375°, 7.0313° to 175.7813°, 8.4375° to 177.1875°, 9.8438° to 178.5938°, each in steps of 11.25°). For each trial, we thus obtained 800 samples for each of the 16 Mahalanobis distances. The distances were averaged across the samples of each trial and ordered as a function of orientation difference. The resulting 'similarity profile' was summarised into a single value (i.e., 'decoding accuracy') by computing the cosine vector mean of the similarity profile [18], in which a positive value suggests a higher pattern similarity between similar orientations than between dissimilar orientations. The approach was the same for the reanalysis of [17].

We also repeated the above analysis iteratively for a subset of electrodes in a searchlight analysis across all 61 electrodes. In each iteration, the 'current' as well as the closest two neighbouring electrodes were included in the analysis (similar as in [50]). The freely available MATLAB extension fieldtrip [51] was used to visualise the decoding topographies. Note that the topographies were flipped, such that the left represents the ipsilateral and the right the contralateral side relative to stimulus presentation side.

### Orientation code generalisation

To test cross-generalisation between impulses, instead of training and testing within the same time window, the train folds were taken from the impulse 1 epoch, and the test fold from the

impulse 2 epoch, and vice versa. The analysis was otherwise exactly as described above using 8-fold cross-validation with separate trials in each fold.

To test cross-generalisation between presented cued locations (i.e., whether the cued item was previously presented on the left or on the right), the classifier was similarly trained on trials in which the cued item was presented on the left and tested on trials in which the cued item was presented on the right, and vice versa. Since left and right trials were independent trial sets, cross-validation does not apply. However, to ensure a balanced training set, the number of trials of each orientation bin were nevertheless equalised by subsampling (as described above), and this approach was repeated 100 times.

The cross-generalisation of the orientation code between impulse onsets in [17] was tested with the same analyses as the location cross-generalisation described in the paragraph above: the classifier was trained on the early-onset condition and tested on the late-onset condition, and vice versa, while making sure that the training set is balanced through random subsampling.

## Impulse/time and location decoding

To decode the difference of the evoked neural responses between impulses, we used a leave-one-out approach. The Mahalanobis distances between the signals from a single trial from both impulse epochs and the average signal of all other trials of each impulse epoch were computed. The covariance matrix was computed by concatenating the trials of each impulse (excluding the left-out trial). The average difference of same-impulse distances was subsequently subtracted from different impulse distances, such that a positive distance difference indicates more similarity between same than different impulses. To convert the distance difference into trial-wise decoding accuracy, positive distance differences were simply converted into 'hits' (1) and negative into 'misses' (0). The percentages of correctly classified impulses were subsequently compared to chance performance (50%).

The presentation side and impulse onset (in [17]) was decoded using 8-fold cross-validation, in which the distance difference between different and same location/onset was computed for each trial, which was then converted to 'hits' and 'misses'.

## Visualisation of the spatial, temporal, and orientation code

To explore and visualise the relationship between the location or impulse/time code and the orientation code in state space (see Fig 1A for different predictions), we used classical MDS of the Mahalanobis distances between the average signal of trials belonging to one of four orientation bins (0˚ to 45˚, 45˚ to 90˚, 90˚ to 135˚, 135˚ to 180˚) and location (left/right) or time (impulse 1/impulse2).

For the visualisation of the code across impulse/time, distances were computed separately for left and right trials, before taking the average. Within each orientation bin, the data of half of the trials were taken from impulse 1, and the data of the other half from impulse 2 (determined randomly). The number of trials within each orientation of each impulse were equalised through random subsampling before averaging. The Mahalanobis distances between both orientation and impulses were then computed using the covariance matrix estimated from all trials of both impulses. This was repeated 100 times (for each side), randomly subsampling and splitting trials between impulses each time and then taking the average across all iterations.

For the visualisation of the code across space, the data of each trial were first averaged across impulses. The number of trials of orientation bins (same as above) of each location was equalised through random subsampling. The Mahalanobis distances of the average of each bin

within each location condition were computed using covariance estimated from all left and right trials. This was repeated 100 times before taking the average across all iterations.

For the code across impulse onset/time visualisation of the data from [17], the same procedure as in the paragraph above was used, but instead of visualising the stimulus code between locations, it was visualised between impulse onsets (−30 ms, +30 ms).

## Relationship between behaviour and the neural representation of the WM item

We were interested in whether imprecise reports that are CW or CCW relative to the actual orientation are accompanied by a corresponding shift of the neural representation in WM (see Fig 1B for model schematics). We used two approaches to test for such a shift (Figs 6A and 7A).

First, the trial-wise pattern similarities as a function of orientation differences (as obtained from the orientation-reconstruction approach described above) were averaged separately for all CW and CCW responses (Fig 6A). Note that CW and CCW responses were defined relative to the median response error within each orientation bin. This ensures a balanced proportion of all orientations in CW and CCW trials, which is necessary to obtain meaningful orientation reconstructions. It furthermore removes the report bias away from cardinal angles in the current experiment (S3 Fig), similar to previous reports of orientation response biases [52], and thus isolates random from systematic report errors.

We used another approach that exaggerates the potential difference between CW and CCW trials and thus might be more sensitive to detect a shift. The data were first divided into CW and CCW trials using the same within-orientation bin approach as described above. The classifier was then trained on CW trials and tested on CCW trials, and vice versa (Fig 7A). The orientation bins in the training set were balanced through random subsampling, and the procedure was repeated 100 times. Given an actual shift in the neural representation, the shift magnitude of the resulting orientation reconstruction of this method should be doubled, since both the testing data and the training data (the reference point) are shifted, but in opposite directions.

To improve orientation reconstruction from the impulse epochs, the classifier was trained on the averaged trials of both impulses but tested separately on each impulse epoch individually. Although training on both impulses improved orientation reconstruction, in particular for the second approach, in which only half of the trials are used for training, the shifts in orientation representations as a function of CW/CCW reports are qualitatively the same when training and testing within each impulse epoch separately (Figs 6 and 7 and S4 Fig).

The resulting similarity profiles for CW and CCW reports were summarised such that a positive/CW shift reflects a shift towards the response. The similarity profile of CCW reports was thus flipped and then averaged with the similarity profile of CW reports. Evidence for a shift in the similarity profile was then computed both at the group and at the participant level. At the group level, the shift magnitude was quantified by averaging the shifted similarity profiles across all participants and then taking the circular mean of the resulting population-level similarity profile. At the participant level, an 'asymmetry score' of each participant's similarity profile was computed by subtracting the pattern similarities of all negative-orientation differences (i.e., −67.5, −45, and −22.5 degrees, which represent orientations away from the response) from all positive-orientation differences (i.e. 67.5, 45, and 22.5 degrees, which represent orientations towards the response). Thus, if the similarity profile is shifted towards the response, then the neural patterns of specific orientations should be more similar to

orientations in the direction of the response error compared to the opposite, resulting in a positive 'asymmetry score'.

## Statistical significance testing

To test for statistical significance of average decoding, we first repeated the decoding analysis in question 1,000 times with randomised condition labels over trials (either orientations, cued location, or impulse), such that the condition labels and the EEG signal were unrelated. The resulting 1,000 values per participant were then transformed into a null distribution of $t$ values, which was used to perform a $t$ test against chance performance with a significance threshold of $p = 0.05$. Note that tests of within-condition decoding (within presentation location, impulse/onset) were one-sided, since only positive decoding is plausible in those cases, whereas tests of cross-generalisation between conditions were two-sided, since negative decoding is theoretically plausible in those cases.

Comparisons of decodability between conditions/items were tested for statistical significance by subtracting the 1,000 values of each 'null' decoder from another before computing the null distribution of difference $t$ values. All difference tests were two-sided.

A null distribution for the 'asymmetry score' towards the response was obtained by randomising the report errors within each orientation bin, meaning that trials within each bin were randomly labelled CW and CCW. In the case of the 'report-dependent averaging of similarity profiles' (Fig 6A), report errors were randomised with respect to the trial-wise similarity profiles of the orientation decoder output 1,000 times. In the case of the 'response-dependent training and testing' (Fig 7A), report errors were random with respect to the EEG signal before training the orientation decoder on randomly labelled 'CCW' trials and testing it on the other trials that are randomly labelled 'CW' (and vice versa) 1,000 times. These randomly averaged similarity profiles were then used in both cases to obtain a null distribution of 'asymmetry score' $t$ values, which in turn was used to perform a $t$ test on the 'asymmetry scores' against zero.

The circular mean of the shifted average similarity profile at the group level was tested against 0. The similarity profile of each participant was flipped left to right with 0.5 probability, such that a participant's positively shifted similarity profile would then be negatively shifted, before computing the circular mean of the resulting similarity profile averaged over all participants 100,000 times. The resulting null distribution was used to obtain the $p$-value by calculating the proportion of permuted similarity profiles with circular means more positive than the observed group-level circular mean.

All tests of similarity profile shift (asymmetry score and circular mean) were one-sided, since we expected the shift of the neural representation of the orientation to be towards the response.

For visualisation, we computed the 95% CIs by bootstrapping the data in question 100,000 times.

## Supporting information

**S1 Fig. Full cross-temporal decoding matrix of the orientation of the cued item between impulses.** Black bars indicate the presentation times of the impulses. Continuous EEG data from posterior channels (see Methods) were baselined relative to impulse 1 (−200 to 0 ms), smoothed with a gaussian smoothing kernel (SD = 16 ms), and down-sampled to 100 Hz. The classifier (the same as described in the Methods) was then trained and tested on all possible time point–by–time point combinations. Data available at osf.io/cn8zf. EEG, electroencephalography; SD, standard deviation.
(TIF)

**S2 Fig. Cross-generalisation of coding scheme between impulse onsets in reanalyses of [17].** (A) Visualisation of orientation and impulse-onset code in state space. The third dimension discriminates between impulse onsets. The first and second dimensions code the orientation space in both impulses. (B) Trial-wise accuracy (%) of impulse-onset decoding. (C) Orientation decoding within each impulse onset (blue) and orientation code cross-generalising between impulse onsets (green). Error shadings and error bars are 95% CI of the mean. Asterisks indicate significant decoding accuracies or cross-generalisation ($p < 0.05$). Data available at osf.io/cn8zf.
(TIF)

**S3 Fig. Report bias of orientations.** Participants showed a bias, exaggerating the tilt of oblique orientations, manifesting itself as a repulsion form the cardinal axes (0 and 90 degrees; left), similar to previous reports [52]. To ensure an unbiased estimate of a possible shift in our analysis and to isolate random from systematic errors, the report bias was removed by subtracting the median error within 11.25-degree orientation bins (middle). By removing orientation-specific error, the resulting error distribution is narrower (right). Clockwise and counterclockwise reports were defined as positive and negative reports relative to this 'adjusted', unbiased report error. Data available at osf.io/cn8zf.
(TIF)

**S4 Fig. Within-impulse training and testing to estimate drift.** (A) Response-dependent averaging of trial-wise similarity profiles (Fig 6A). Shift towards response: Impulse 1: $p = 0.492$ (circular mean), $p = 0.500$ (asymmetry score); Impulse 2: $p = 0.022$ (circular mean), $p = 0.020$ (asymmetry score), one-sided. (B) Response-dependent training and testing (Fig 7A). Shift towards response: Impulse 1: $p = 0.545$ (circular mean), $p = 0.525$ (asymmetry score); Impulse 2: $p = 0.009$ (circular mean), $p = 0.004$ (asymmetry score), one-sided. Same convention as Figs 6B, 6C, 7B and 7C. Data available at osf.io/cn8zf.
(TIF)

## Acknowledgments

We would like to thank N. E. Myers and D. Trübutschek for helpful comments.

## Author Contributions

**Conceptualization:** Michael J. Wolff, Janina Jochim, Elkan G. Akyürek, Timothy J. Buschman, Mark G. Stokes.

**Data curation:** Michael J. Wolff, Janina Jochim.

**Formal analysis:** Michael J. Wolff.

**Funding acquisition:** Elkan G. Akyürek, Mark G. Stokes.

**Investigation:** Michael J. Wolff, Janina Jochim, Elkan G. Akyürek, Timothy J. Buschman, Mark G. Stokes.

**Methodology:** Michael J. Wolff, Timothy J. Buschman, Mark G. Stokes.

**Project administration:** Janina Jochim, Mark G. Stokes.

**Resources:** Janina Jochim.

**Software:** Michael J. Wolff.

**Supervision:** Mark G. Stokes.

**Validation:** Michael J. Wolff.

**Visualization:** Michael J. Wolff, Timothy J. Buschman, Mark G. Stokes.

**Writing – original draft:** Michael J. Wolff, Mark G. Stokes.

**Writing – review & editing:** Michael J. Wolff, Janina Jochim, Elkan G. Akyürek, Timothy J. Buschman, Mark G. Stokes.

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
