## [Editor Report · Decision Letter 0]

5 Aug 2019

Dear Mark, 

Thank you for submitting your manuscript entitled "Drifting codes within a stable coding scheme for working memory" for consideration as a Research Article by PLOS Biology.

Your manuscript has now been evaluated by the PLOS Biology editorial staff, as well as by an Academic Editor with relevant expertise, and I am writing to let you know that we would like to send your submission out for external peer review.

*Please be aware that, due to the voluntary nature of our reviewers and academic editors, manuscripts may be subject to delays during the holiday season. Thank you for your patience.*

Please re-submit your manuscript within two working days, i.e. by Aug 07 2019 11:59PM.

Kind regards,

Gabriel Gasque, Ph.D.,

Senior Editor

PLOS Biology

---

## [Decision Letter · Decision Letter 1]

7 Oct 2019

Dear Mark,

Thank you very much for submitting your manuscript "Drifting codes within a stable coding scheme for working memory" for consideration as a Research Article at PLOS Biology. Your manuscript has been evaluated by the PLOS Biology editors, by an Academic Editor with relevant expertise, and by three independent reviewers. You will note that reviewer 1, Edward Ester, has signed his comments. Please accept my sincere apologies for the delay in sending the decision below to you.

In light of the reviews (below), we will not be able to accept the current version of the manuscript, but we would welcome resubmission of a much-revised version that takes into account the reviewers' comments. We cannot make any decision about publication until we have seen the revised manuscript and your response to the reviewers' comments. Your revised manuscript is also likely to be sent for further evaluation by the reviewers.

Your revisions should address the specific points made by each reviewer. Having discussed these with the Academic Editor, we think you should pay special attention to the following points:

Reviewer 1

Requests various control analyses, including a more rigorous analysis to test or control for potential effects of eye movements on decoding performance.

Also, rationale for choice of 10ms bin size and then for collapsing across time as a variable to increase statistical power.

Reviewer 2

Addresses various analysis and controls, including clarifying whether mean activity was subtracted separately for training and test data sets

Clarify definition of dynamic code and what constitutes a dynamic code

Reviewer 3

Requests various control analyses and greater clarification of rationale for various analyses.

The non-parametric permutation test that was used was not standard, and did not adequately model variability at the level of individual participant data. We think you should randomly shuffle the labels of the trial labels at model estimation and use this null decoder to recover orientation reconstructions, and compare resulting metrics of decoding accuracy across participants from a distribution of results between a null and intact decoder. 

Rationale for varying the number of orientation bins across different analyses. Also, is post hoc selection of analyses that reach statistical significance an issue here?

How do the results look without the orientation smoothing kernel applied?

----------

Please submit a file detailing your responses to the editorial requests and a point-by-point response to all of the reviewers' comments that indicates the changes you have made to the manuscript. In addition to a clean copy of the manuscript, please upload a 'track-changes' version of your manuscript that specifies the edits made. This should be uploaded as a "Related" file type. You should also cite any additional relevant literature that has been published since the original submission and mention any additional citations in your response. 

Before you revise your manuscript, please review the following PLOS policy and formatting requirements checklist PDF: http://journals.plos.org/plosbiology/s/file?id=9411/plos-biology-formatting-checklist.pdf. It is helpful if you format your revision according to our requirements - should your paper subsequently be accepted, this will save time at the acceptance stage.

Please note that as a condition of publication PLOS' data policy (http://journals.plos.org/plosbiology/s/data-availability) requires that you make available all data used to draw the conclusions arrived at in your manuscript. If you have not already done so, you must include any data used in your manuscript either in appropriate repositories, within the body of the manuscript, or as supporting information (N.B. this includes any numerical values that were used to generate graphs, histograms etc.). For an example see here: http://www.plosbiology.org/article/info%3Adoi%2F10.1371%2Fjournal.pbio.1001908#s5.

For manuscripts submitted on or after 1st July 2019, we require the original, uncropped and minimally adjusted images supporting all blot and gel results reported in an article's figures or Supporting Information files. We will require these files before a manuscript can be accepted so please prepare them now, if you have not already uploaded them. Please carefully read our guidelines for how to prepare and upload this data: https://journals.plos.org/plosbiology/s/figures#loc-blot-and-gel-reporting-requirements.

Upon resubmission, the editors will assess your revision and if the editors and Academic Editor feel that the revised manuscript remains appropriate for the journal, we will send the manuscript for re-review. We aim to consult the same Academic Editor and reviewers for revised manuscripts but may consult others if needed.

We expect to receive your revised manuscript within two months. Please email us (plosbiology@plos.org) to discuss this if you have any questions or concerns, or would like to request an extension. At this stage, your manuscript remains formally under active consideration at our journal; please notify us by email if you do not wish to submit a revision and instead wish to pursue publication elsewhere, so that we may end consideration of the manuscript at PLOS Biology.

When you are ready to submit a revised version of your manuscript, please go to https://www.editorialmanager.com/pbiology/ and log in as an Author. Click the link labelled 'Submissions Needing Revision' where you will find your submission record. 

Sincerely,

Gabriel Gasque, Ph.D., 

Senior Editor

PLOS Biology

Reviewer remarks:

Reviewer #1: This is an interesting study that leverages a clever manipulation to get at an important yet tricky issue. The analyses are appropriate and vigorous, and the authors' complementary re-analysis of earlier published work suggests that the key findings are robust. I have a few relatively minor comments that the authors should be able to address in a revision. In no particular order or priority: 

1. Possible typo on p. 9? The text states that the authors focused on windows spanning 100-400 ms after each impulse, then sort the data into 10 ms bins. That should produce 30 bins per impulse and a total of 17 x 30 = 510 dimensions, not the 50 and 850 described in the paper. 

2. In collapsing across time the authors' stated goal is to improve statistical power. So, why bother with the 10 ms bins? The authors don't provide a justification for selecting that bin size (nor can I think of one), so I naturally wonder whether key findings generalize if the data are sorted in a different (arbitrary) way. For example, do the authors get the same thing if they average data across the whole 300 ms post-impulse epoch (which would presumably maximize statistical power)? What happens with larger bin sizes (50 ms, 100 ms, etc)?

3. Eye movements can contribute to EEG decoding (Mostert et al eNeuro 2018; Quax et al NIMG 2019), and small but robust eye position biases have been found in retrocue tasks like the one used here (van Ede et al NHB 2019). The authors went to great pains to minimize effects of eye movements by manually rejecting trials with visible EOG displacements and applying ICA to remove other signals from the residual data set. Still, small-but-consistent eye movements could contribute to decoding performance, for example if a subject moved his or her eyes along the axis of the retrocued orientation. Successful cross-hemifield decoding argues against biases in eye position if one assumes that those biases reflect a mixture of stimulus location and orientation, but a more formal and perhaps compelling approach would be to regress residual horizontal EOG activity onto the orientation of the cued stimulus whether there's any systematic relationship between eye position and voltage - hopefully not!

These issues aside I think this is an interesting paper that provides important empirical support for an issue that heretofore has only been studied computationally. 

Signed,

Edward Ester

Reviewer #2: Summary

In this paper the authors apply multivariate pattern analysis to EEG recordings in order to gain insight into how information is stored in working memory. In particular, the authors use a paradigm where they flash blank stimuli in order to assess what information is stored in working memory about a previously shown grating. The results show several interesting

findings including that: 1) initially the information is contained in the contralateral early visual cortex and that during maintenance the information shifts more bilaterally, 2) information seems to only be present about the attended stimulus, 3) the information about stimulus orientation is contained in a generally stable code along with information about time, 4) information seems to shift during the course of a trial in a way that is correlated with behavioral errors.

While I have not kept up with the literature on examining dynamic coding using EEG, so I am not familiar with the authors’ previous work or the novelty of this work, from my perspective this paper does seem very nice, with a good experiment design and analysis methods that can extract subtle patterns from noisy EEG signals. Some of the data analysis methods are a bit ad hoc so it might be useful to include some more standard analyses to verify the findings and potentially give additional insights, at least as supplemental figures. There are also a few statements in the paper that do not seem correct, but overall this paper seems very interesting. 

Major issues

Line 47 (and also line 463): There are some dynamic codes that can be read out using a simple set of linear weights, as is typically done for stable codes, so saying that dynamic codes “necessarily entails a more complex readout strategy” is not really correct. For example, imagine that there is a perfect synfire chain of neurons where each neuron fires only to one stimulus at one particular point in time. Then a fixed set of weights would work perfectly well for decoding since having the irrelevant weight outside of a neuron’s time-stimulus selective window wouldn’t matter since the neuron would never fire at that time, while they would contribute positively to predictions in their stimulus-time window. However, this type of coding scheme would be considering dynamic by most people (e.g., it would have a strong diagonal pattern in a temporal cross decoding analysis). 

Many areas do show this type of sparse sequence coding (see Hahnloser, Kozhevnikov and Fee, Nature 2002), and even for less sparsely firing areas, dynamic neurons often fire at much higher rates during window when they are selective so that their lower firing rates outside these windows would not contribute much noise to any simple linear decoder. There have been some decoding analyses showing that using a fixed set of weights on a dynamic code still can extract most of the information present (see Sreenivasan and D’Esposito Nat Rev Neuro 2019 and Meyers J Neurophys 2018). Changing this sentence and citing these other reviews would be useful. 

Line 174: When the mean activity was subtracted, was this done separately on the training set and then applied to the test set? If not, then data leakage is possible. If it was done separately, it would be good to state this. Similarly, on line 180, the averaging was done separately for the training and test sets right? Stating this and that none of the same data is in the training and test sets would be good (overall I found the description around line 180 to be confusing). Finally, it would be good to state that separate trials are used around line 211. 

Figure 4. It would be interesting to see the full temporal cross decoding matrix for this and perhaps some of the other results as well, either as a main figure or a supplemental figure. This would be informative to see how long the information is maintained for and how stable the code is, even if there is only information at the time of the pulses. Also, why is “decoding accuracy used for impulse decoding and “pattern similarity” used for orientation analysis. It would be nice to see both these measures for all types of information as a main or supplemental figure. 

Line 356 (and elsewhere). When you say the errorbars are 95% CIs, are these CIs calculated over the different participants? I don’t think how these CIs are created are described in the methods section. 

Minor issues

Figure 1. For part A, it would be good to label the axes, and to put the labels for the bar chart below the corresponding bars (or use a color key) to make it clearer what the bars correspond to. Mentioning that for the middle plot that some features are stable and others are dynamic would be useful (and perhaps labeling which is which). For part B it would be good to label the axes and include a color bar.

Line 153: Technically applying ICA across all the data first before doing the cross-validation analyses could lead to data leakage, so if you are manually rejecting eye artifacts anyway, leaving this step out could be better, although in all likelihood this is fine (particularly because if it was data leakage then it would be possible to decoding the unattended stimulus as well). 

Line 215: You mean to say when the left stimulus was attended, right? My understanding of the design is that the stimuli were always presented bilaterally. It might also be good to say that fixation is inherently controlled by the fact that eye movements would cause artifacts, since I am assuming there is no other fixation control. 

Line 279: Would this analysis be more sensitive if the magnitude of the mistakes was taken into account? 

Line 310: Before describing the results, it would be useful to reiterate the some of the rationale of the study. 

Line 315: with a p-value around 0.057 it could just be a lack of power for the reason you don’t see information about the uncued item. The fact that the curve looks tuned to the correct orientation and that there is no statistically significant difference between the cued and uncued location also suggests this is the case. 

Figure 3 is very nice. I don’t think you need to explain what a boxplot is in the caption in this or any of the other figures since it is a pretty common type of plot. 

Paragraph 365-375. I don’t think one would expect too many dynamics in a short window of 100 ms difference. I think this paragraph can be removed. 

Line 379. It is a very interesting finding that the code in EEG is stable. It would be good to mention that this is at a more global scale of brain networks that EEG picks up and doesn’t necessarily mean there is a stable pattern at the level of neurons. 

Line 384. It is interesting that the orientation information appears to be a lower level visual signal that don’t generalize across locations. Perhaps orientation information would generalize across position in PFC but can’t be picked up by EEG due to the fact that there is less retinotopy in PFC? 

Line 466. The amount of dynamics might depend on the task. Remembering a stimulus orientation is a simple task (no transformation of information needed) which might explain why the code is stable. 

Line 494: Might be better to say that drifts along the orientation dimension are “correlated” with recall errors rather than “predicted” since you didn’t try to predict errors on individual trials (although it would be neat if you tried to do that).

Reviewer #3: Wolff and colleagues characterized the dynamics of impulse-evoked EEG response patterns across working memory (WM) delay intervals. This group has previously pioneered a multivariate analysis approach that uses sensory-evoked responses during WM delay intervals to assay feature-selective WM representations at different points in time during the trial. The general approach involves requiring participants encode a visual stimulus (oriented grating) into visual WM over a brief delay interval, during which time a transient visual stimulus is presented to ‘ping’ the brain, evoking a content-selective pattern of activity. In the present study, Wolff et al wanted to assay, within a single delay period, whether the WM representation is stable. Does it look the same at different points in time, or does it transform dynamically? They applied their impulse response decoding technique to recover feature-selective orientation representations throughout the delay period. By applying generalization methods, they demonstrate that orientation is represented in a similar manner early and late in the delay period, but is not represented the same way when it originates from the left vs right side of the screen. Furthermore, despite the similarity in representations suggested by the generalizability of the decoder, it was still possible to recover temporal information (which part of the delay period the impulse occurred during), which provides evidence that the WM representation is stable in format, but dynamic in an orthogonal dimension. Finally, to further probe how WM representations might be dynamic over the delay period, the authors sorted trials based on the type of behavioral error (clockwise vs counterclockwise). Decoded orientations tracked behavioral errors, which accumulated over the delay – there was a bias in the decoded orientation at later periods of the delay, but not during encoding or the early part of the delay. 

The study is quite convincing, and I anticipate the results will be of broad interest to a general neuroscience audience. The authors do a particularly nice job of preempting some initial concerns I had about interpretation of their results in their discussion (e.g., how a drifting code can square with an activity-silent coding framework). I have some small concerns with some analysis choices, and with the statistical treatment of results, which I describe below.

1. In some analyses (the ‘orientation reconstruction’ analysis), the authors use 16 orientation bins (e.g., Fig. 3), while for the MDS visualizations (e.g., Fig. 4A), they use 4 – are similar conclusions supported with a more granular form of the analysis – maybe with 8 or 16 bins?

2. Similarly, the authors smooth the ‘orientation reconstructions’ with a kernel derived from a cosine raised to a high power. How do results look without the smoothing kernel applied? I know the quantification should not change, as the smoothing kernel will not impact the decoding accuracy or circular bias measures reported, but I’m curious what the data look like when no smoothing is used.

3. Previous studies have used induced alpha power to assay spatial working memory representations (e.g., Foster et al, 2016; Ester et al, 2018). I understand the authors are not employing a spatial task here, but I’m curious if they’ve tried using alpha power induced throughout the delay interval (not just in response to the impulses) when analyzing WM representations. It’s certainly possible participants are co-opting a spatial strategy, which would invoke an alpha-related mechanism. (Alternatively, because the stimuli were presented laterally, alpha could carry information about the spatial origin of the WM stimulus throughout the delay; e.g., Foster et al, 2017, Current Biology). In any case, I’m curious how the stability of an alpha-based code, if there is one in this task, compares with the orthogonally-dynamic code in the impulse-evoked responses (and/or, how the authors consider such induced alpha WM codes in context of these results). 

4. I applaud the authors for employing non-parametric permutation-based statistics throughout the manuscript. However, the tests they use do not seem to take into account the variability in the sample when evaluating significance. For example, when evaluating whether decoding accuracy was above chance, they randomly flipped the sign of the average decoding accuracy (or, equivalently, the orientation reconstruction) and computed a new average across participants 100,000 times, providing a ‘null’ distribution of average decoding accuracies against which to compare the actual value. However, comparing an average to a distribution of averages isn’t extremely meaningful without also considering the variability across participants. As an example, a typical inferential statistic, like a T-test, computes an estimate of difference between two means, divided by a measure of the variability across the sample. Under normality, etc, assumptions, a T-score can be used to estimate a p-value. The p-values computed in this report seem to consider only a null distribution of mean values, rather than a null distribution of mean values compared to their variability. So, I’d suggest instead generating something like a null T distribution (compute a T-score across participants for each permutation iteration, and compare a final T score to the null T distribution rather than the parametric T distribution). Additionally, for evaluating whether decoding accuracy is above chance, it’s often useful to estimate a decode w/ a “null” training set – shuffle trial labels at model estimation and use this null decoder to recover orientation reconstructions, and compare resulting metrics of decoding accuracy across participants from a distribution of results between a null and intact decoder. (together with a variability-based metric, like null T distribution of shuffled decoding accuracies vs decoding accuracy computed using intact labels). Similar adjustments to statistical methods can be made to the other analyses employed.

5. At the beginning of the Results (lines 311-312), the authors conclude that they recovered “parametric information about the presented orientations”, with similar conclusions reached throughout the Results. However, I’m not sure any of the statistical analyses employed conclusively demonstrate ‘parametric’ coding. The visualized reconstructions have been smoothed by a cosine kernel, and the statistical tests seem to be using the “decoding accuracy” metric, which only identifies significant information overall, not the ‘shape’ of that information. (for example, that metric would be reliably positive if there was one large value at the center orientation, and equally-small values at all other orientations). The authors should clarify the basis for these statements, and/or add a direct test of the ‘smoothness’ of the representations.

6. Line 326-327, the authors conclude a difference between contralateral vs ipsilateral sites in Fig. 3 for the Memory Array, but not for the two impulses , but this doesn’t seem to be supported by a statistical test.

Minor:

1. Fig. 1A – what are the dimensions meant to be in these sketches? Neurons? 

2. Line 110 – I don’t believe an orientation error can be greater than 90 degrees

3. A few places (like line 409), the authors describe their decoding results as “orientation tuning curve(s)” – I’m not sure this is the best nomenclature to use, since tuning curves/function most typically describes the feature-dependent response properties of recorded neurons. This is instead a somewhat different result – a function describing the relative multivariate distance between an observed activity pattern and lots of other prototype activity patterns.

4. Fig. 6C, 7C – throughout the other figures, the authors use box plots to elegantly and transparently report their decoding results, but these figures use more typical means/error bars – can box plots be used here too? Or, due to the circular nature of the data, is this not possible?

---

## [Editor Report · Decision Letter 2]

30 Dec 2019

Dear Mark,

Thank you for submitting your revised Research Article entitled "Drifting codes within a stable coding scheme for working memory" for publication in PLOS Biology. I have now discussed your revision with the Academic Editor, and I am delighted to let you know that we're now editorially satisfied with your manuscript. 

However before we can formally accept your paper and consider it "in press", we also need to ensure that your article conforms to our guidelines. A member of our team will be in touch shortly with a set of requests. As we can't proceed until these requirements are met, your swift response will help prevent delays to publication. Please also make sure to address the data and other policy-related requests noted at the end of this email.

*Copyediting*

*Published Peer Review History*

*Early Version*

*Submitting Your Revision*

Sincerely,

Gabriel Gasque, Ph.D., 

Senior Editor

PLOS Biology

ETHICS STATEMENT:

The Ethics Statements in the submission form and Methods section of your manuscript should match verbatim. Please ensure that any changes are made to both versions.

-- Please indicate if your approved protocols by the Central University Research Ethics Committee of the University of Oxford adhered to the Declaration of Helsinki or any other national or international guidelines. 

-- Please include within your manuscript the ID number of the approved protocols. 

DATA POLICY:

-- Please ensure that the figure legends in your manuscript include information on where the underlying data can be found, and ensure your supplemental data file/s has a legend.

---

## [Editor Report · Decision Letter 3]

12 Feb 2020

Dear Dr. Stokes,

On behalf of my colleagues and the Academic Editor, Dr. Frank Tong, I am pleased to inform you that we will be delighted to publish your Research Article in PLOS Biology. 

Early Version

PRESS 

Kind regards,

Krystal Farmer

Development Editor

PLOS Biology

on behalf of

Gabriel Gasque,

Senior Editor

PLOS Biology